# On-chip temporal focusing of elastic waves in a phononic crystal waveguide

M. Kurosu[1,2], D. Hatanaka[1], K. Onomitsu[1] & H. Yamaguchi[1,2]

The ability to manipulate acoustic and elastic waveforms in continuous media has attracted significant research interest and is crucial for practical applications ranging from biological imaging to material characterization. Although several spatial focusing techniques have been developed, these systems require sophisticated resonant structures with narrow bandwidth, which limit their practical applications. Here we demonstrate temporal pulse manipulation in a dispersive one-dimensional phononic crystal waveguide, which enables the temporal control of ultrasonic wave propagation. On-chip pulse focusing is realized at a desired time and position with chirped input pulses that agree perfectly with the theoretical prediction. Moreover, traveling four-wave mixing experiments are implemented, providing a platform on which to realize novel nonlinear phenomena in the system. Incorporating this dispersive pulse engineering scheme into nonlinear phononic crystal architecture opens up the possibility of investigating novel phenomena such as phononic solitons.

[1] NTT Basic Research Laboratories, NTT Corporation, Atsugi-shi, Kanagawa 243-0198, Japan. [2] Department of Physics, Tohoku University, Sendai 980-8578, Japan. Correspondence and requests for materials should be addressed to M.K. (email: kurosu_megumi_s5@lab.ntt.co.jp) or to D.H. (email: hatanaka.daiki@lab.ntt.co.jp)

Acoustic and elastic ultrasonic waves have been widely utilized in various applications and are especially important as a tool for nonintrusive sensing in such areas as biological imaging and defense systems[1–4]. The capabilities of these systems, for example, imaging resolution and accuracy, are determined by the spatial size and energy density of generated waves. To improve these capabilities, several spatial focusing techniques have been proposed where engineered structures such as the phased arrays, negative-index materials, and acoustic metamaterials enable acoustic and elastic waves to be focused in tiny spatial regions[5–8]. Although such structures have been used to demonstrate effects such as nearly diffraction limited focusing and super-focusing in spatial domain[5,6], they often require locally resonant structures and anisotropic property of the band structure, which results in narrow operation bandwidth and thus limits their practical use.

In addition to spatial control, temporal manipulation of ultrasonic vibration is important for modern communications, signal processing, and sensing technologies. Especially, chip scale techniques will play a significant role in the manipulation of nanomaterials and lab-on-a-chip technology. Recently, the effects of surface strain on the adsorption and diffusion properties of molecules or atoms on graphene or other materials have attracted great attention, because an understanding of these effects will have a great impact on future applications ranging from selective molecular synthesis to drug delivery[9,10]. The pulse manipulation technique controls mechanical strain in the time domain, which enables molecular synthesis at a desired position and time in a tunable manner. The effect of dispersion in a medium is the key to temporal pulse manipulation. Temporal pulse manipulation can be demonstrated by utilizing the concept of space-time duality, which describes the mathematical equivalence between paraxial-beam diffraction and dispersive pulse broadening[11,12]. Importantly, this temporal focusing method can be applied to an arbitrary dispersive material which imparts quadratic time-varying phase shift.

In this study, by introducing the approach in the field of phononics, we demonstrate on-chip temporal pulse manipulation in a one-dimensional (1D) phononic crystal waveguide (PnC WG)[13,14] which is constructed by using nanoelectromechanical

systems (NEMS) technology. Ultrasonic waves traveling through the WG experience pulse broadening due to the group velocity dispersion (GVD) effect. This GVD effect can be used to compensate for the frequency modulation of the initial pulse that leads to temporal focusing. In addition, a four-wave mixing (FWM) experiment using traveling elastic waves is demonstrated for the first time, and it clarifies the availability of nonlinear phenomena in elastic pulses such as phonon solitons in the PnC WG. By incorporating the excellent mechanical properties of NEMS such as high-quality factors, integrability, and nonlinearity into this device[15–18], the ability to temporally focus the traveling ultrasonic waves will open up the possibility of developing an ultrashort and highly intense ultrasonic pulse generator, which will be useful for practical applications as was the case for optical laser systems[19–21], and for investigating nonlinear phononic phenomena.

## Results

**Phononic crystal waveguide.** The PnC WG consists of a 1 mm long membrane made from a GaAs/AlGaAs heterostructure as shown in Fig. 1a, where periodically spaced air holes with a pitch of 8 μm are formed along the WG that can be used to suspend the 22 μm wide membrane by selectively etching an underlaying $Al_{0.65}Ga_{0.35}As$ layer. The application of an alternating voltage to an electrode located at both edges of the WG induces ultrasonic vibrations caused by the piezoelectric effect. The resultant vibrations travel down the WG and are detected by an optical interferometer.

Figure 1b, c shows the experimental transmission spectrum of the device and the corresponding band structure calculated by using a finite element method (FEM) simulation (COMSOL Multiphysics). The ultrasonic vibrations are observed at 3.5–7.5 MHz owing to the presence of the first phonon band, where the vibrations propagate in the WG and are reflected at the clamped edges, thus resulting in the generation of equidistant Fabry–Perot peaks in the frequency domain. On the other hand, ultrasonic waves around 8 MHz experience Bragg reflection from the periodic air holes, giving rise to a phonon bandgap, and this prevents them from propagating in the WG[13,14]. Above the

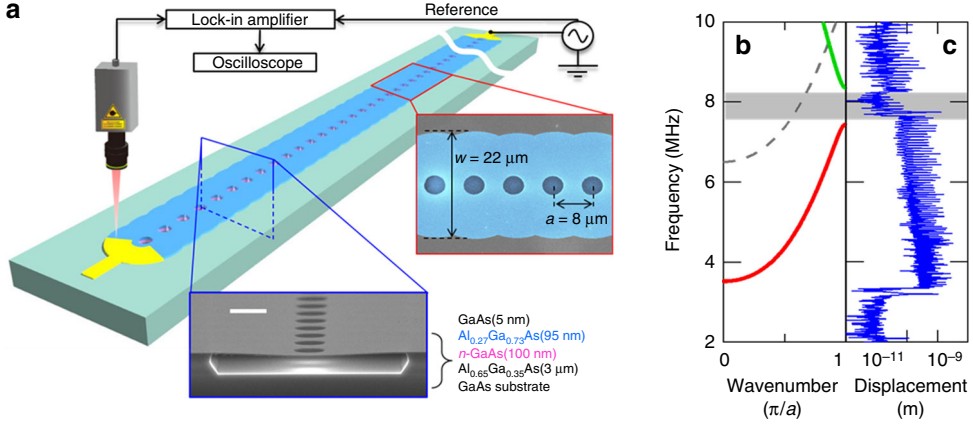

**Fig. 1** 1D phononic crystal waveguide. **a** A schematic of the PnC WG and the measurement setup. The bottom inset shows SEM image of the cross-section of the device which is composed of a GaAs/AlGaAs heterostructure fabricated by selectively etching the $Al_{0.65}Ga_{0.35}As$ layer, scale bar is 5 μm. The periodic structures are determined by the WG width of 22 μm and the hole pitch of 8 μm as shown in the right inset (false colored SEM image). The nanomechanical vibrations are piezoelectrically excited at the right edge and detected at the left edge with a laser Doppler interferometer at room temperature and in a high vacuum ($2 \times 10^{-4}$ Pa). **b** The FEM simulated dispersion relation of the PnC WG. The internal stress between the GaAs and $Al_{0.27}Ga_{0.73}As$ layers is included in the simulation[34]. The dashed line indicates the second phonon band, which does not contribute to the transmission due to the piezoelectric transducer electrodes being located at the nodal position of the mode in this band. **c** The experimental transmission spectrum, which is measured at the left edge by employing continuous excitation with 1.0 $V_{rms}$ from the right edge

bandgap, there is a new phonon branch that again allows the vibrations to be guided. In the experiments described below, we focus on the first phonon branch to investigate the temporal dynamics of the ultrasonic vibrations in the device.

**Dispersion effect on ultrasonic temporal waveform.** A transverse deflection $z(x, t)$ traveling in a 1D PnC WG can be described by Euler–Bernoulli equations as[22]

$$EI \frac{\partial^4 z(x,t)}{\partial x^4} + \rho S \frac{\partial^2 z(x,t)}{\partial t^2} + b_1 z(x,t) + b_3 z^3(x,t) = 0, \quad (1)$$

where $E$ is the Young's modulus, $S$ and $I$ are the area and moment of inertia of the cross-section, $\rho$ is the density of the WG per unit length, and $b_1$ and $b_3$ are the elastic coefficients of the WG. To solve Eq. (1), the slowly varying amplitude of traveling wave is assumed that the envelope of a traveling vibration pulse centered around wavenumber $k$ and angular frequency $\omega$ varies slowly in temporal and spatial domain on the moving-frame with a group velocity,

$$z(x, t) = A(x, t) \exp[i(kx - \omega t)]. \quad (2)$$

We introduce the linear loss term $\alpha$, then $A(x, t)$ satisfies the following equation[22]:

$$i \frac{\partial A}{\partial x} = -\frac{i\alpha}{2} A + \frac{k_2}{2} \frac{\partial^2 A}{\partial t^2} - ik_1 \frac{\partial A}{\partial t} - \xi A^2 \overline{A}, \quad (3)$$

where $k_1 = \frac{\partial k}{\partial \omega} \equiv v_g^{-1}$ is the inverse of the group velocity, $k_2 = \frac{\partial^2 k}{\partial \omega^2}$ is the GVD coefficient, and $\xi$ is a nonlinear parameter. These terms are determined by the material and geometric parameters of the WG in Eq. (1). Thus, the dynamics of wave propagation based on the Euler–Bernoulli equation corresponds to a nonlinear Schrödinger equation (NLSE), which is used to describe the dynamics of an optical wave propagating in a dispersive medium[23–25]. By neglecting the nonlinear term in Eq. (3) and using the new time coordinate $T$ moving with the group velocity $v_g$,

$$T = t - \frac{x}{v_g} = t - k_1 x. \quad (4)$$

At the same time we introduce a normalized amplitude $U(x, T)$,

$$A(x, T) = A_0 \exp(-\alpha x / 2) U(x, T), \quad (5)$$

where $A_0$ is the peak amplitude of the input pulse, and the equation can be simplified as

$$i \frac{\partial U}{\partial x} = \frac{k_2}{2} \frac{\partial^2 U}{\partial T^2}. \quad (6)$$

Interestingly, Eq. (6) is mathematically equivalent to the paraxial wave equation that governs the diffraction of CW light. If we assume a Gaussian pulse as an input, the normalized amplitude at distance $x$ is given by[23]

$$U(x, T) = \frac{T_0}{(T_0^2 - ik_2 x)^{1/2}} \exp\left(-\frac{T^2}{2(T_0^2 - ik_2 x)}\right), \quad (7)$$

where $T_0$ is the half-width at the $1/e$-intensity point, and its output width $T_1$ is written as

$$T_1(x) = T_0 \sqrt{1 + \left(\frac{|k_2|x}{T_0^2}\right)^2}. \quad (8)$$

Thus an ultrasonic temporal waveform traveling down a dispersive medium depends on the absolute value of the GVD

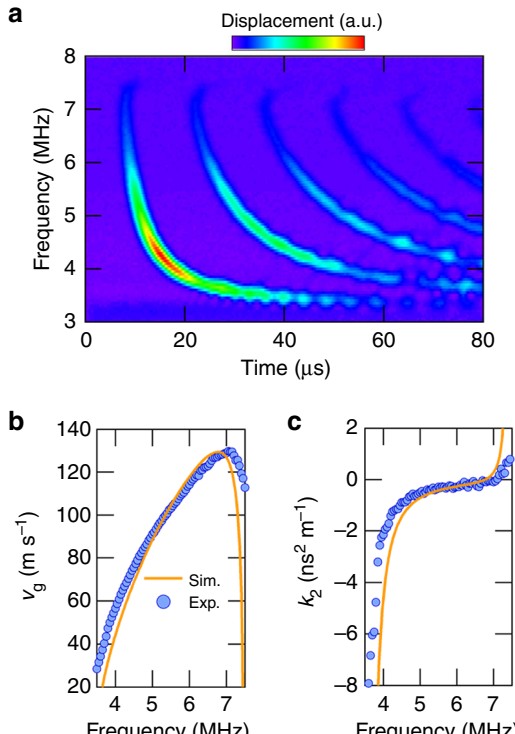

**Fig. 2** Fundamental properties of the PnC WG. **a** The temporal response of ultrasonic wave propagations measured at the left edge of the WG when exciting the input Gaussian pulse with $T_0 = 1.2$ μs and an amplitude of 1.0 $V_{rms}$ from the right edge. The shape of the applied wave is the same as that in Fig. 3a. The leftmost curve consists of waves propagating 1 mm, the second curve consists of waves propagating 3 mm (round-trip after 1 mm propagation), and so on. **b**, **c** The frequency dependence of the group velocity $v_g$ and the GVD coefficient $k_2$, respectively, where the experimental and FEM simulated results are denoted as circles and solid lines, respectively. The experimental values are obtained as follows. From the fitting of the leftmost Gaussian pulses, the center position of the wave $t_1$ is determined, then $v_g$ is calculated from $v_g = L/t_1$, where $L$ is the propagation length. $k_2$ is calculated from $k_2 = \frac{\partial}{\partial \omega} \frac{1}{v_g}$, where $\omega$ is the angular frequency

coefficient that enables the temporal pulse width to be broadened further due to the larger dispersion.

To elucidate the temporal characteristics of this device experimentally, ultrasonic vibrations are measured at the left edge by exciting them with a Gaussian pulse with $T_0 = 1.2$ μs from the right edge as shown in Fig. 2a. This time-of-flight measurement also enables the group velocity $v_g$ and the GVD coefficient $k_2$ to be estimated as a function of excitation frequency as shown in Fig. 2b, c, respectively. The experimental $v_g$ value increases with increasing frequency, i.e. anomalous dispersion ($k_2 < 0$) in the band except near the bandgap where there is a large reduction in $v_g$, i.e. normal dispersion ($k_2 > 0$), because of the decreased slope of the band. In addition, the temporal waveform of the pulse around the band edges is distorted and becomes asymmetric with an oscillation near the trailing edge caused by a contribution from the third-order dispersion effect (Supplementary Note 1). These experimental results can be well reproduced by the calculation results obtained with FEM-simulated band structures.

The temporal pulse widths can also be estimated from the time-of-flight measurement in Fig. 2a by fitting a Gaussian envelope to the output waveforms. Figure 3b, c shows the output pulse widths at various excitation frequencies when exciting with Gaussian pulses with $T_0 = 1.2$ and 0.7 μs, respectively, and

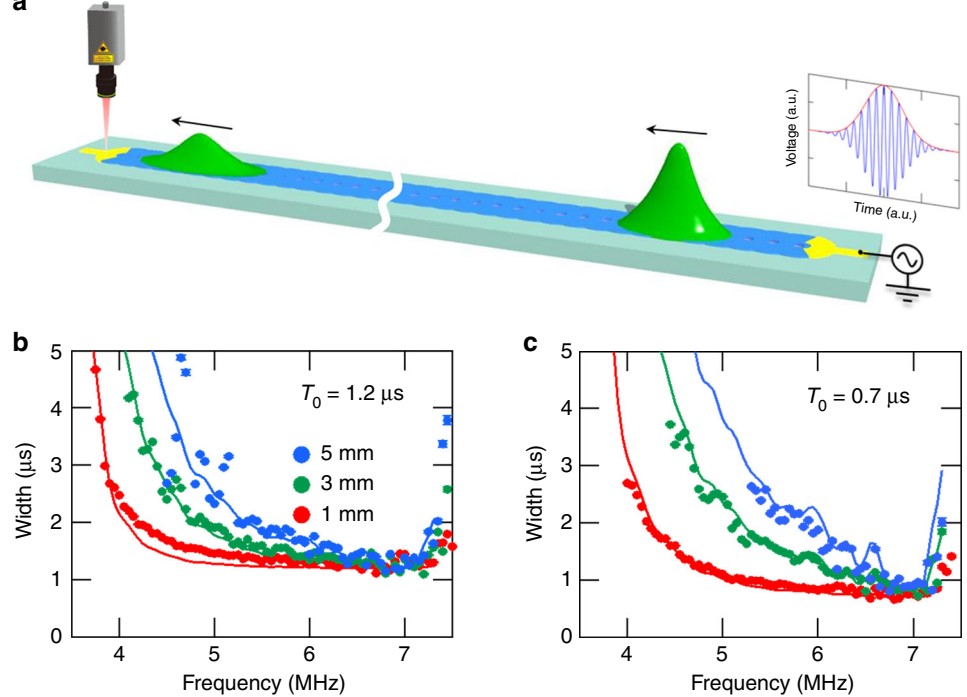

**Fig. 3** Pulse broadening caused by the group velocity dispersion effect. **a** A schematic showing the pulse broadening due to the GVD effect when injecting an unchirped Gaussian pulse. **b**, **c** The frequency dependence of the temporal width in the output waveform measured at propagation distances $x = 1$ (red), 3 (green), and 5 (blue) mm by exciting Gaussian input pulses with $T_0 = 1.2$ and $0.7\,\mu s$, respectively. The solid line indicates the theoretical results estimated by substituting the experimentally obtained GVD coefficient $k_2$ in Fig. 2c into Eq. (8). Error bars denote one standard deviation

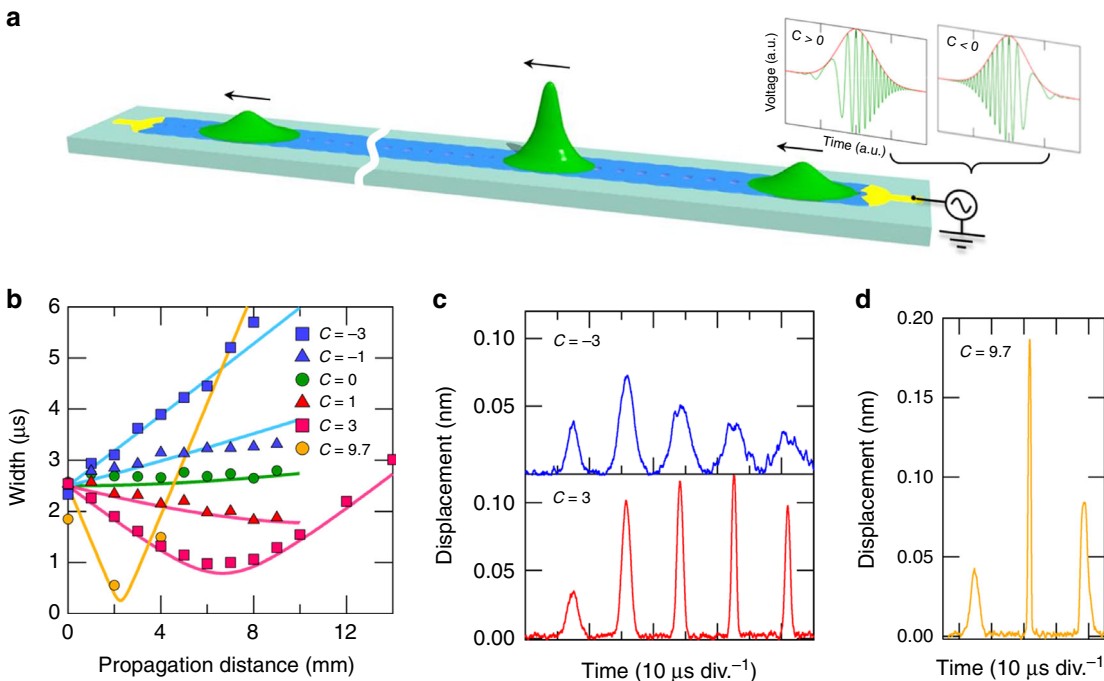

**Fig. 4** Temporal focusing of ultrasonic pulses. **a** A schematic showing the pulse focusing caused by the GVD effect when injecting a chirped Gaussian pulse. **b** Temporal pulse width as a function of propagation distance when exciting a 2.5 μs chirped pulse with a center frequency of 5.8 MHz and various chirp parameters. The GVD coefficient $k_2$ at this frequency is $-0.28\,\mathrm{ns}^2\,\mathrm{m}^{-1}$ in Fig. 2c, indicating an anomalous dispersion regime. **c**, **d** Temporal evolution of the ultrasonic wave when exciting a chirped Gaussian pulse with a chirp parameter $C = \pm3$ and 9.7 and measuring it at the right edge. The observed wave packets except for the first are associated waves composed of the incident and reflected waves at the WG edge, which increase the peak amplitude nearly twofold while maintaining the pulse width

measuring them at distances $x = 1$, 3, and 5 mm (see Fig. 3a). As theoretically predicted, the pulse widths are greatly increased as the frequency approaches the band edges where $|k_2|$ is large and the GVD-induced pulse broadening becomes distinct with increasing propagation distance as shown in Fig. 3b, c[26]. In particular, shortening the input pulse from $T_0 = 1.2$ to 0.7 μs, namely spectral broadening, leads to the propagation distance having a significant influence on the output width as shown in Fig. 3c.

**Temporal focusing of an ultrasonic pulse**. In the previous experiment we confirmed that the pulse is broadened during propagation thanks to the GVD effect. In an anomalous (normal) dispersion regime, the high-frequency components of the pulse travel faster (slower) than its low-frequency components, thus allowing the injected unchirped pulse to be frequency chirped and broadened, where the product of the temporal and spectral widths is not transform limited. In general, this effect is unfavorable for efficient wave guiding. However, we utilize this disadvantage to demonstrate temporal focusing, namely the compression and amplification, of the traveling ultrasonic pulse. Here, a frequency-chirped pulse is excited as the input, and frequency modulation within the pulse is compensated for by the GVD via propagation, thus resulting in the pulse being compressed to the lower limit for the pulse width determined by a given spectrum, namely transform limited, and the peak amplitude being amplified. The dynamics of the chirped pulse evolution in the WG can also be described by NLSE, and the output pulse width $T_2$ is given by

$$T_2(x) = T_0 \sqrt{\left(1 + \frac{Ck_2x}{T_0^2}\right)^2 + \left(\frac{k_2x}{T_0^2}\right)^2}. \tag{9}$$

$C$ is a chirp parameter that is positive (negative) when the frequency increases (decreases) linearly from the leading to the trailing edge, and is defined by $\Delta f = \sqrt{(1 + C^2)}/2\pi T_0$ where $\Delta f$ is the spectral half-width at the $1/e$-intensity point. Equation (9) indicates that pulse focusing occurs only when $Ck_2 < 0$, where an up-chirped $C > 0$ (down-chirped $C < 0$) pulse is used as an input in an anomalous (normal) dispersion regime $k_2 < 0$ ($k_2 > 0$).

As an experimental demonstration in this device, an up-chirped pulse is injected in the anomalous dispersion regime between 3.5 and 7 MHz as shown in Fig. 4a. The output pulse width is measured at various distances by exciting different chirped pulses with a center frequency of 5.8 MHz as shown in Fig. 4b–d. The pulse widths with negative $C$ monotonically increase with increasing distance (see Fig. 4b and the upper panel of Fig. 4c), whereas the widths of pulses with a positive $C$ first decrease to the transform limited value, and in turn, increase with distance (see Fig. 4b and the lower panel of Fig. 4c). It should also be noted that a smaller pulse width is observed when we employ a larger absolute positive $C$ value and indeed, strong focusing is realized when $C = 9.7$ where the pulse width is temporally compressed from 1.9 to 0.6 μs, and simultaneously, the spatial width to be compressed from 0.48 to 0.14 mm (Supplementary Note 2). The pulse power is enhanced more than one order of magnitude. Making use of the GVD effect enables a traveling pulse waveform to be engineered that leads to the temporal focusing of ultrasonic waves.

In a conventional spatial lens, the figures of merit such as the compression and amplification factors, and the ability to spatially adjust the focusing position, are mainly determined by the geometric parameters of the system. Therefore, these techniques require sophisticated artificial structures if they are to control waves. In contrast, this pulse manipulation technique enables them to be dynamically designed simply by changing the excitation frequency, input pulse width, and chirp parameters. Although, in this study, these focusing properties are limited by the device structure causing the reflection at both WG edges, and the finite bandwidth of the lock-in amplifier, it is possible to further enhance this focusing effect by modifying the device structure and optimizing the measurement setup.

**Nonlinear effect of the 1D PnC WG**. The dispersion effect is important when demonstrating ultrasonic pulse engineering. On the other hand, in order to induce nonlinear wave phenomena, a nonlinear elastic effect is also required because the emergence of such phenomena results from the balance between the dispersion and nonlinear effects in a medium. Therefore, an investigation of its availability in this PnC WG is particularly desirable.

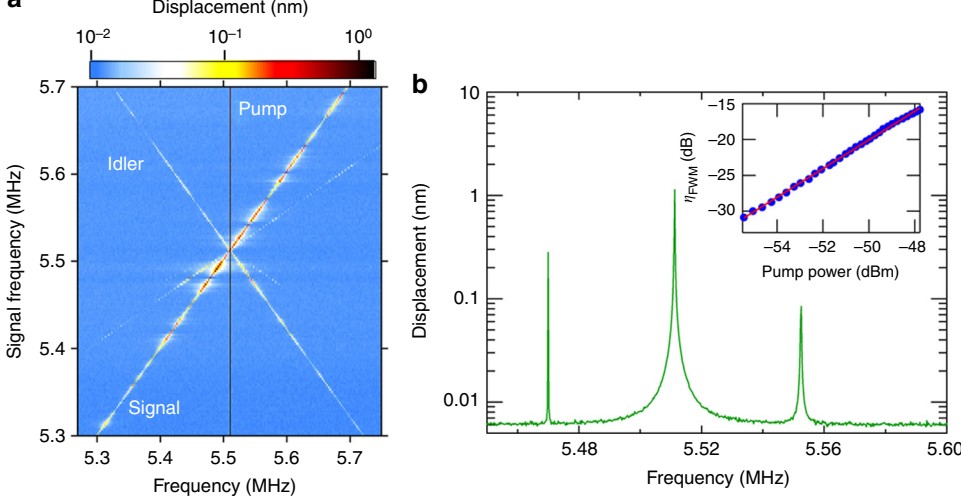

**Fig. 5** Ultrasonic four-wave mixing. **a** The spectral response measured at $L = 9$ mm when exciting the pump at $f_p = 5.5112$ MHz with 2 V$_{rms}$ and sweeping the signal in $f_s = 5.3$–5.7 MHz with 0.5 V$_{rms}$. **b** The spectral response with $f_s = 5.47$ MHz where a new idler wave is generated at $f_i = 5.5524$ MHz. The inset shows the conversion efficiency $\eta_{FWM}$ as a function of the pump input power. The blue circles and red line show the experimental results and theoretical fitting from Eq. (10), respectively

FWM is a well-known nonlinear phenomenon in optics. It is based on a third-order nonlinear process that allows a propagating wave (signal) to be frequency converted to a new wave (idler) via highly intense wave excitation (pump). To the best of our knowledge, FWM has not been demonstrated for traveling ultrasonic waves in a waveguide because the generation of a sufficiently intense wave has been challenging. To induce the nonlinear effect, interdigital transducer (IDT) electrodes are introduced at the edge of the WG where 50 IDT fingers are arrayed with a pitch of 20 μm (Supplementary Note 3). The IDT can excite highly intense waves in the device and allows us to access the third-order elastic nonlinearity of the device. Additionally, the WG length is increased to 33 mm, and this suppresses the reflection of the ultrasonic wave and the Fabry–Perot resonance. Thus, this IDT-embedded PnC WG makes it possible to observe the nonlinear effect caused by the propagating ultrasonic waves.

By investigating the conversion efficiency of idler waves to signal waves with the pump excitation power, the nonlinear parameter $\xi$ of this device can be estimated. The spectral response of the device is measured at the propagation length $L = 9$ mm when exciting the pump wave at $f_p = 5.5112$ MHz and sweeping the signal frequency $f_s = 5.3 – 5.7$ MHz as shown in Fig. 5a. Idler waves are continuously generated between 5.7224 and 5.3224 MHz and the frequency is always at $f_i = 2f_p - f_s$. Figure 5b indicates the measured spectrum when exciting the signal wave at $f_s = 5.47$ MHz, where the idler wave is created at $f_i = 2f_p - f_s = 5.5524$ MHz, thus indicating that the energy conservation is satisfied. From such an FWM spectrum, the conversion efficiency $\eta_{FWM}(L)$ can be estimated from the power ratio of the idler output $P_i(L)$ to the signal input $P_s(0)$, and it is given by[27,28]

$$\eta_{FWM}(L) = \frac{P_i(L)}{P_s(0)} = \left(\xi P_p(0) L_{eff}\right)^2 \left(\frac{\sinh(gL)}{gL}\right)^2 \exp(-\alpha L), \quad (10)$$

where $P_p(0)$ is the input power of the pump (Supplementary Note 4), $L_{eff} = (1 - \exp(-\alpha L))/\alpha$ is the effective length, and $\alpha$ is the propagation loss. $g$ is the parametric gain and is expressed by

$$g = \sqrt{\left(\xi P_p(0) \frac{L_{eff}}{L}\right)^2 - \left(\frac{\Delta k L + 2\xi P_p(0) L_{eff}}{2L}\right)^2}, \quad (11)$$

where $\Delta k = \frac{1}{2} k_2 (\omega_p)(\omega_p - \omega_s)^2$, and $\omega_p$ and $\omega_s$ are the angular frequencies of the pump and signal, respectively. In Eq. (10), the conversion efficiency increases quadratically with increasing pump power, and the dependence is confirmed experimentally as shown in the inset of Fig. 5b. By calculating the dispersion relation with FEM and performing a time-of-flight measurement in this device, $\sinh(gL)/gL$ and $\alpha$ are obtained as almost unity and 0.35 dB mm$^{-1}$, respectively. As a result, fitting Eq. (10) to the experimental results enables the nonlinear parameter to be roughly estimated as $\xi \sim 2 \times 10^9$ W$^{-1}$ m$^{-1}$.

## Discussion

The estimated nonlinear parameter allows us to provide the requirement for the observation of soliton. On the assumption of a temporal pulse width $T_0 = 2.5$ μs, GVD coefficient $k_2 = -5 \times 10^{-10}$ s$^2$ m$^{-1}$, and a peak displacement of 10 nm, second-order solitons will emerge at a propagation distance $L = 20$ mm[23]. They are reasonable values with which to experimentally realize a soliton with the current device structure and experimental configuration. Additionally, this nonlinear wave phenomenon can be efficiently induced by incorporating materials with a high piezoelectric constant such as AlN and ZnO into the device and operating at cryogenic temperatures. Thus, the 1D PnC WG allows both dispersion and nonlinear effects to be efficiently

managed, and this will provide a powerful tool with which to manipulate ultrasonic waveforms via nonlinear wave phenomena.

Improving the manipulation of ultrasonic vibration on a chip is important for modern communications and sensing technologies. Compared to the widely used micromechanical resonators, waveguide structures have several advantages, such as a wide frequency range, high-speed operation, and low energy consumption. The ability to adjust the timing and location of focusing elastic waves on a chip will develop new technologies, allowing not only the manipulation of nanomaterials but also quantum devices to be selectively driven by locally confined strains[29,30]. Furthermore, this temporal focusing technique can be applied in two-dimensional phononic crystals and metamaterials where elastic wave propagations are directionally controlled based on the anisotropic property of 2D band structures[31–33], to create new means to manipulate elastic waves on chip.

In conclusion, we have demonstrated an on-chip ultrasonic pulse manipulation in a 1D PnC WG. The dispersion effect determined by the periodic geometry of the device induces frequency chirp in a Gaussian input pulse during propagation. This phenomenon also allows the traveling wave to be temporally focused, and this can be controlled by changing the input chirp parameters. Moreover, the nonlinearity of the device is investigated from FWM measurements, which offers the possibility of realizing nonlinear phenomena such as solitons. This on-chip temporal pulse control of elastic waves can be utilized to develop novel chip based technologies.

## Methods

**Measurement setup**. Nanomechanical vibrations were excited in the PnC WG by applying an amplitude modulated alternating voltage from a signal generator (NF Wavefactory 1974 and 1968), and were measured with an He–Ne laser Doppler interferometer (NEOARK MLD-230V-200-NN). In the spectral measurements in Figs. 1c and 5a, b, the electrical output from the interferometer was measured with a vector signal analyzer (HP89410A). In the temporal measurements in Figs. 2a, 3b, c and 4c, d, the electrical output was first filtered with a lock-in amplifier (Zurich Instruments UHFLI), and then measured with an oscilloscope (Agilent DSO6014A). The spectral bandwidth of the low-pass filter in the lock-in amplifier was set at 470 kHz, which is equivalent to a time constant of 102.6 ns.

**Data availability**. The data that support the findings of this study are available from the corresponding author on request.

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

## Acknowledgements

We are grateful to Y. Ishikawa for growing the heterostructure. This work is partly supported by a MEXT Grant-in-Aid for Scientific Research on Innovative Areas "Science of hybrid quantum systems" (Grant No. JP15H05869 and JP15K21727).

## Author contributions

M.K. and D.H. performed the measurements and data analysis. M.K. and D.H. fabricated the sample and K.O. co-fabricated the GaAs/AlGaAs heterostructure. M.K. and D.H. wrote the paper and H.Y. planned the project.

## Additional information

**Competing interests:** The authors declare no competing interests.

