## [Peer Review File(PDF 694 kb) · Nature Communications]

Reviewers' comments:

Reviewer #1 (Remarks to the Author):

The authors use a one-dimensional waveguide based on mechanical resonators in order to demonstrate a phononic time lens. In two previous papers (Refs. 16,17), the authors reported the fabrication of such devices and the measurement of transmission spectra. These two papers generated considerable interest because these works were at the boundary between two research fields, nanomechanical resonators and phononics. Here, the authors used these devices to demonstrate compression and amplification of traveling phonon pulses. The measurements are spectacular and are well described by the model introduced in the manuscript. These results are promising for the study of nonlinear phononic phenomena, such as phononic solitons. I highly recommend publication.

Here are a few minor comments

- 1- It would be useful for readers to show an example of the shape of the wave (displacement versus time) in Fig. 2.
- 2- It would be nice if the authors explain how v_g and k_2 are extracted from Fig. 2a
- 3- Why is the displacement largest at 4 MHz in Fig. 2a?
- 4- The authors may explain the origin of the different curves in Fig. 2a.
- 5- In Fig. 1b, the scale of the displacement axis is difficult to understand.

Reviewer #2 (Remarks to the Author):

The main claim of the paper is the observation of pulse compression at a specific position along a phononic waveguide. The waveguide is claimed to be for phonons, but it would be more accurate to say it is for ultrasonic waves in a structured beam. Wave propagation is in the linear regime, and although the authors include nonlinearities in their equations, nonlinear effects do not appear in the measurements. The measurements are illustrative of the propagation of pulses with a Gaussian envelope and a tailored spectral phase. Actually, the physical situation is analogous to radar technology (and acoustic variants such as sonar or ultrasound imaging) or ultrashort laser pulse technology. The effect of second order (group delay dispersion) and third order dispersion is very well known and understood. Calling dispersion compensation a time lens does not create new physics in my view. For these reasons, I don't consider the results to be specifically original, though they are likely correct.

The interest of the study is justified by potential applications to biological imaging or material characterization. In ultrasonic imaging and non destructive evaluation, pulse imaging techniques are standard (think of echographic systems for medical imaging). Furthermore, it is claimed that 'a highly intense strain field' can be created: do the authors classify 1 nm of displacement at a few MHz as such? The study is claimed to enable the investigation of novel nonlinear phononic phenomena such as phononic solitons and rogue waves, but nothing is proved in these directions and I doubt the strain levels obtained can be sufficient.

In summary, I can't support publication of the paper as it is presented.

Reviewers' comments:

Reviewer #1 (Remarks to the Author):

The authors use a one-dimensional waveguide based on mechanical resonators in order to demonstrate a phononic time lens. In two previous papers (Refs. 16,17), the authors reported the fabrication of such devices and the measurement of transmission spectra. These two papers generated considerable interest because these works were at the boundary between two research fields, nanomechanical resonators and phononics. Here, the authors used these devices to demonstrate compression and amplification of traveling phonon pulses. The measurements are spectacular and are well described by the model introduced in the manuscript. These results are promising for the study of nonlinear phononic phenomena, such as phononic solitons. I highly recommend publication.

Here are a few minor comments

- 1- It would be useful for readers to show an example of the shape of the wave (displacement versus time) in Fig. 2.
- 2- It would be nice if the authors explain how v_g and k_2 are extracted from Fig. 2a
- 3- Why is the displacement largest at 4 MHz in Fig. 2a?
- 4- The authors may explain the origin of the different curves in Fig. 2a.
- 5- In Fig. 1b, the scale of the displacement axis is difficult to understand.

Reviewer #2 (Remarks to the Author):

The main claim of the paper is the observation of pulse compression at a specific position along a phononic waveguide. The waveguide is claimed to be for phonons, but it would be more accurate to say it is for ultrasonic waves in a structured beam. Wave propagation is in the linear regime, and although the authors include nonlinearities in their equations, nonlinear effects do not appear in the measurements. The measurements are illustrative of the propagation of pulses with a Gaussian envelope and a tailored spectral phase. Actually, the physical situation is analogous to radar technology (and acoustic variants such as sonar or ultrasound imaging) or ultrashort laser

pulse technology. The effect of second order (group delay dispersion) and third order dispersion is very well known and understood. Calling dispersion compensation a time lens does not create new physics in my view. For these reasons, I don't consider the results to be specifically original, though they are likely correct.

The interest of the study is justified by potential applications to biological imaging or material characterization. In ultrasonic imaging and non destructive evaluation, pulse imaging techniques are standard (think of echographic systems for medical imaging). Furthermore, it is claimed that 'a highly intense strain field' can be created: do the authors classify 1 nm of displacement at a few MHz as such? The study is claimed to enable the investigation of novel nonlinear phononic phenomena such as phononic solitons and rogue waves, but nothing is proved in these directions and I doubt the strain levels obtained can be sufficient.

In summary, I can't support publication of the paper as it is presented.

We would like to thank both reviewers for carefully reading our manuscript and providing variable comments. We performed additional experiments by fabricating the device with a newly designed transducer structure and wrote a new section “Nonlinear effect of the 1D PnC WG” to clarify the role of nonlinearity based on the experiments. We believe that this major revision will help the reviewers and readers to appreciate the importance and originality of our study. We provide detailed answers to the reviewers' individual comments below.

To Reviewer #1

=====

The authors use a one-dimensional waveguide based on mechanical resonators in order to demonstrate a phononic time lens. In two previous papers (Refs. 16,17), the authors reported the fabrication of such devices and the measurement of transmission spectra. These two papers generated

considerable interest because these works were at the boundary between two research fields, nanomechanical resonators and phononics. Here, the authors used these devices to demonstrate compression and amplification of traveling phonon pulses. The measurements are spectacular and are well described by the model introduced in the manuscript. These results are promising for the study of nonlinear phononic phenomena, such as phononic solitons. I highly recommend publication.

=====

We are grateful to the reviewer for the very positive comments on our study. We have included an additional section to emphasize the importance of our experiments as regards the study of nonlinear phononic phenomena.

=====

Q1-1. It would be useful for readers to show an example of the shape of the wave (displacement versus time) in Fig. 2.

=====

A1-1. We thank the reviewer for the suggestion. Actually, the shape of the wave in Fig. 2 is the same as that in Fig 3. Therefore, we have added this to the figure caption of Fig. 2 as mentioned below.

The shape of the applied wave is the same as that in Fig. 3(a).

=====

Q1-2. It would be nice if the authors explain how v_g and k_2 are extracted from Fig. 2a

=====

A1-2. Again, we thank the reviewer for the suggestion. We have added an explanation of how to extract v_g and k_2 to the figure caption of Fig. 2 as mentioned below.

The experimental values were obtained as follows. The center position of the wave t_1 is determined by fitting Gaussian pulses, then v_g is calculated from $v_g = \frac{L}{t_1}$, where L , is the propagation length. k_2 is calculated from $k_2 =$

$\frac{\partial}{\partial \omega} \left(\frac{1}{v_g} \right)$, where ω is angular frequency.

=====

Q1-3. Why is the displacement largest at 4 MHz in Fig. 2a?

=====

A1-3. As the reviewer mentioned, the displacement depends on the excitation frequency. We estimate that the electrode shape matches the mode shape at around 4 MHz, so the electrode can efficiently transduce voltage to mechanical vibration.

=====

Q1-4. The authors may explain the origin of the different curves in Fig. 2a.

=====

A1-4. We have modified our description of the origin of the different curves in the caption of Fig. 2 as mentioned below.

The leftmost curve represents waves which propagate 1 mm, and the other curves represent reflected waves.

==>

The leftmost curve consists of waves propagating 1 mm, the second curve consists of waves propagating 3 mm (round-trip after 1 mm propagation), and so on.

In response to Q1-1, Q1-2 and Q1-4, we have revised the figure caption of Fig. 2 as below.

(a) The temporal response of ultrasonic wave propagations measured at the left edge of the WG when exciting the input Gaussian pulse with $T_0 = 1.2 \mu\text{s}$ and an amplitude of $1.0 V_{\text{rms}}$ from the right edge. The shape of the applied wave is the same as that in Fig. 3(a). The leftmost curve consists of waves propagating 1 mm, the second curve consists of waves propagating 3 mm (round-trip after 1 mm-propagation), and so on.

(b), (c) The frequency dependence of the group velocity v_g and the GVD coefficient k_2 respectively, where the experimental and FEM simulated results are denoted as circles and solid lines, respectively. The experimental

values are obtained as follows. From the fitting of the leftmost Gaussian pulses, the center position of the wave t_1 is determined, then v_g is calculated from $v_g = \frac{L}{t_1}$, where L , is the propagation length. k_2 is calculated from $k_2 = \frac{\partial}{\partial \omega} \left(\frac{1}{v_g} \right)$, where ω is the angular frequency.

=====

Q1-5. In Fig. 1b, the scale of the displacement axis is difficult to understand.

=====

A1-5. We have changed the displacement axis in Fig. 1b.

To Reviewer #2

We are grateful to hear the reviewer's opinion of our manuscript and greatly appreciate her/his feedback. We have responded to the comments below.

=====

Q2-1. The main claim of the paper is the observation of pulse compression at a specific position along a phononic waveguide. The waveguide is claimed to be for phonons, but it would be more accurate to say it is for ultrasonic waves in a structured beam.

=====

A2-1. We agree with reviewer #2. Vibration is not a quantum phenomenon and so we should use the term ultrasonic instead of phonon. Therefore, we have changed "phonon wave" to "ultrasonic wave" in the manuscript. However, the term phononic crystal is generally used in the field of phononics even in the classical regime. In order to encourage researchers in the field to find the new aspects in the application of phononic crystals, we would like to retain the term "phononic crystal".

=====

Q2-2. Wave propagation is in the linear regime, and although the authors

include nonlinearities in their equations, nonlinear effects do not appear in the measurements.

=====

A2-2. We agree with the reviewer that we did not include experiments on nonlinear effects in the previous manuscript. To confirm nonlinear effects in propagating waves experimentally, we fabricated a new device with an interdigital transducer electrode, which enabled us to excite large amplitude waves in the device. Although acoustic four-wave mixing has already been observed in resonator and cavity structures, it is much more difficult to observe it in an acoustic waveguide without resonance. We clearly demonstrated four-wave mixing in the new device, and estimated the nonlinear parameter. From the nonlinear parameter, we estimated the condition for observing a second order soliton in our device. We believe that this result will pave the way to the manipulation of ultrasonic waves such as electromagnetic waves or light waves. Experimental results for acoustic four-wave mixing are described in a new section entitled “Nonlinear effect of the 1D PnC WG”

=====

Q2-3. The measurements are illustrative of the propagation of pulses with a Gaussian envelope and a tailored spectral phase. Actually, the physical situation is analogous to radar technology (and acoustic variants such as sonar or ultrasound imaging) or ultrashort laser pulse technology. The effect of second order (group delay dispersion) and third order dispersion is very well known and understood. Calling dispersion compensation a time lens does not create new physics in my view. For these reasons, I don't consider the results to be specifically original, though they are likely correct.

=====

A2-3. As the reviewer mentioned, the dispersion compensation technique is widely used in radar detection systems, ultrashort laser pulse generation and the acoustic field because this technique is very important for practical applications. However, to the best of our knowledge, there have been few demonstrations of dispersion compensation in a phononic crystal, which enables the precise control of the dispersion relation. In our measurements, the experimental results are perfectly matched to a theoretical prediction on

a chip scale. This fact indicates that this technique can be applied to chip-scale devices. Therefore, this demonstration is important for practical applications such as surface acoustic wave filters and ultrasonic sensors. Furthermore, as we demonstrated, the nonlinear effect is investigated in the device, which indicates that the device has the potential to be a test-bed for fundamental research. Although research on nonlinearity in fluids or air has progressed, nonlinearity in solid ultrasonic waveguides has yet to be fully investigated. We believe there is plenty of room for improvement, and we took the first step toward to manipulating ultrasonic waves by combining dispersion effects and nonlinear effects.

=====

Q2-4. The interest of the study is justified by potential applications to biological imaging or material characterization. In ultrasonic imaging and non-destructive evaluation, pulse imaging techniques are standard (think of echographic systems for medical imaging). Furthermore, it is claimed that 'a highly intense strain field' can be created: do the authors classify 1 nm of displacement at a few MHz as such? The study is claimed to enable the investigation of novel nonlinear phononic phenomena such as phononic solitons and rogue waves, but nothing is proved in these directions and I doubt the strain levels obtained can be sufficient.

=====

A2-4. We thank the reviewer for this remark. Pulse imaging techniques are powerful tools for biological imaging or material characterization because optical imaging is difficult in these systems. In addition, our device has the potential to be used as an on-chip device, a test-bed for investigating nonlinearity or chaos in micro/nanomechanics and spectroscopy.

As we already mentioned in A2-2 and described in the new section entitled “Nonlinear effect of the 1D PnC WG”, we demonstrated the four-wave mixing of a propagating wave and estimated the nonlinear parameter. When estimating the realization of solitons, an ultrasonic pulse with displacement 10 nm is sufficient to measure a soliton. This prediction indicates that pulses with nanometer scale displacement can be used to investigate nonlinear phenomena. Actually, we observed a few nanometer displacements in

ultrasonic waves using the 1D PnC WG with IDT electrode. Therefore, the device has the ability to create a highly intense strain field and has the potential to realize solitons.

Reviewers' comments:

Reviewer #1 (Remarks to the Author):

The authors addressed my comments/questions and those of reviewer 2 in a satisfactory manner. The authors added new measurements demonstrating a nonlinear phononic behaviour in their device, which strengthens the importance of the work. I recommend publication.

Reviewer #2 (Remarks to the Author):

The authors have rather deeply modified a part of their paper in response to my comments, by introducing a new sample and providing nonlinear measurements. Regarding the remaining of the paper, changes are minimal. As a side comment, nonlinear effects in MEMS structures are very common and the results and discussion on nonlinear effects is too short.

As I wrote earlier, the results in the paper are probably correct, but are not sufficiently novel to justify publication in this journal. The new nonlinear measurements are a slight progress, but the result still falls too short in displacement amplitude to allow for the observation of solitons, which is the goal the authors set for their study. The conclusion that solitons should be observed provided amplitudes are further increased is very weakly supported by the available evidence.

As I already mentioned in my first review, I find some choices of words very overselling, or even misleading. Is there any physical reason to speak of a 'time lens' when all the literature talks of pulse compression at some position along a waveguide because of dispersion compensation? This is exactly what fiber optics technologists do every day, for instance, but do they talk of 'time lenses'?

I am strongly against publication.

Reviewers' comments:

Reviewer #1 (Remarks to the Author):

The authors addressed my comments/questions and those of reviewer 2 in a satisfactory manner. The authors added new measurements demonstrating a nonlinear phononic behaviour in their device, which strengthens the importance of the work. I recommend publication.

Reviewer #2 (Remarks to the Author):

The authors have rather deeply modified a part of their paper in response to my comments, by introducing a new sample and providing nonlinear measurements. Regarding the remaining of the paper, changes are minimal. As a side comment, nonlinear effects in MEMS structures are very common and the results and discussion on nonlinear effects is too short.

As I wrote earlier, the results in the paper are probably correct, but are not sufficiently novel to justify publication in this journal. The new nonlinear measurements are a slight progress, but the result still falls too short in displacement amplitude to allow for the observation of solitons, which is the goal the authors set for their study. The conclusion that solitons should be observed provided amplitudes are further increased is very weakly supported by the available evidence.

As I already mentioned in my first review, I find some choices of words very overselling, or even misleading. Is there any physical reason to speak of a 'time lens' when all the literature talks of pulse compression at some position along a waveguide because of dispersion compensation? This is exactly what fiber optics technologists do every day, for instance, but do they talk of 'time lenses'?

I am strongly against publication.

Authors' Response to Reviewers' comments:

To Reviewer #1

The authors addressed my comments/questions and those of reviewer 2 in a satisfactory manner. The authors added new measurements demonstrating a nonlinear phononic behavior in their device, which strengthens the importance of the work. I recommend publication.

No response necessary.

To Reviewer #2

The authors have rather deeply modified a part of their paper in response to my comments, by introducing a new sample and providing nonlinear measurements. Regarding the remaining of the paper, changes are minimal. As a side comment, nonlinear effects in MEMS structures are very common and the results and discussion on nonlinear effects is too short.

The authors agree that the nonlinear phenomenon was not described in detail in the main text. This was because the major significance of this paper is not the nonlinearity but the on-chip temporal control of acoustic pulses and because too much detailed description of nonlinearity would defocus the importance. Instead, we included more details on the nonlinearity in supplementary information in last revision.

As I wrote earlier, the results in the paper are probably correct, but are not sufficiently novel to justify publication in this journal.

We thank reviewer #2 for agreeing on the correctness of our paper, but we disagree with his/her comments on the novelty of our paper. We want emphasis that the most significant achievement in this study is *on-chip* and *device-based* demonstration of temporal pulse control, not the proof of feasibility for observing acoustic solitons.

The new nonlinear measurements are a slight progress, but the result still falls too short in displacement amplitude to allow for the observation of solitons, which is the goal the authors

set for their study. The conclusion that solitons should be observed provided amplitudes are further increased is very weakly supported by the available evidence.

We measured the nonlinearity of the device, which enables us to predict conditions for realizing solitons, namely a pulse with 10-nm amplitude and 2.5- μ s width is necessary for observing acoustic solitons. Although waves with 1-nm amplitude are generated by IDT, we cannot create ultrasonic solitons due to the limitation of the material properties of GaAs. However, the important point here is that our scheme can be applied to any material. The electromechanical coupling coefficient, K^2 , is one of the most important figures of merit of transducers. While K^2 is 0.07 % in GaAs, it is about 1~ 8 % in AlN and ZnO, which indicates much larger amplitude waves can be generated in them. We will definitely be able to observe a soliton by using different material for the transducers.

As I already mentioned in my first review, I find some choices of words very overselling, or even misleading. Is there any physical reason to speak of a 'time lens' when all the literature talks of pulse compression at some position along a waveguide because of dispersion compensation? This is exactly what fiber optics technologists do every day, for instance, but do they talk of 'time lenses'?

We consider that a time lens can be realized with any technique that imposes a linear chirp on the pulse. We created input chirped pulses with a signal generator and compressed them by using a dispersive phononic waveguide. However, as reviewer #2 mentioned, a time lens is usually implemented with an electro-optic phase modulator, by mixing the signal pulse with a chirped pulse in a nonlinear crystal and by cross-phase modulation. Therefore, we replaced “time lens” with “temporal pulse focusing”. We emphasize that the physics and results in the manuscript do not change.

Reviewers' comments:

Reviewer #3 (Remarks to the Author):

The paper reports on an original approach for on-chip temporal focusing of elastic waves in a 1D phononic crystal waveguide by pulse manipulation technique. The authors aim the temporal control of ultrasonic wave propagation. The paper also states that the proposed pulse manipulation technique can open the possibility of investigating and implementing new phenomena such as phononic solitons. This paper contains both theoretical/numerical and experimental researches to support the claimed observations and results. Regarding the form, the paper is well written and structured, and the presentation, especially the figures, is of a good quality. Regarding the scientific aspect, the reviewer does think that this paper brings an original approach to deal with on-chip temporal focusing of elastic waves making use of phononic crystal. I like the idea of the paper and how the authors have implemented it. To the opinion of the reviewer, this paper meets the criteria and the standard to be published in Nature Communications, and then the reviewer recommends its publication. However, there are some questions that should be resolved or answered before the potential final publication.

1. The authors are dealing with elastic waves, not acoustic ones. So they must change this both in the title and in the text.
2. In the introduction, the authors briefly mentioned metamaterials as means for wave focusing. Indeed, metamaterials are presenting some unique properties both for focusing and super-focusing (beyond the diffraction limit). The reviewer wants the authors to emphasize more about the add value of their approach in comparison with metamaterials. Complexity and sophistication of some metamaterials is not a sufficient justification to state that they are not simple to fabricate. The literature is full of excellent demonstrations of focusing and super-focusing based on "Metas". Please give some comment on that aspect.
3. Figure 1, the reviewer would like to see a real picture of the sample (Phononic waveguide). A SEM image, for example, could be inserted as a sub-figure in figure 1. It is very important to show the readers the real sample the authors did analyze.
4. The paper is mostly dealing with "temporal focusing" which is very interesting. However, the authors did not give any insight about spatial focusing which is the one of the main properties we look for when we aim any applications regarding wave focusing, lensing and super-focusing. The authors should at least discuss this issue in correlation with the temporal one.
5. The temporal focusing demonstrated by the authors is for 1D elastic wave propagation. Could the authors give some comments on how they can extent this to a 2D configuration ?

Reviewers' comments:

Reviewer #3 (Remarks to the Author):

The paper reports on an original approach for on-chip temporal focusing of elastic waves in a 1D phononic crystal waveguide by pulse manipulation technique. The authors aim the temporal control of ultrasonic wave propagation. The paper also states that the proposed pulse manipulation technique can open the possibility of investigating and implementing new phenomena such as phononic solitons. This paper contains both theoretical/numerical and experimental researches to support the claimed observations and results. Regarding the form, the paper is well written and structured, and the presentation, especially the figures, is of a good quality. Regarding the scientific aspect, the reviewer does think that this paper brings an original approach to deal with on-chip temporal focusing of elastic waves making use of phononic crystal. I like the idea of the paper and how the authors have implemented it. To the opinion of the reviewer, this paper meets the criteria and the standard to be published in Nature Communications, and then the reviewer recommends its publication. However, there are some questions that should be resolved or answered before the potential final publication.

1. The authors are dealing with elastic waves, not acoustic ones. So they must change this both in the title and in the text.

2. In the introduction, the authors briefly mentioned metamaterials as means for wave focusing. Indeed, metamaterials are presenting some unique properties both for focusing and super-focusing (beyond the diffraction limit). The reviewer wants the authors to emphasize more about the add value of their approach in comparison with metamaterials. Complexity and sophistication of some metamaterials is not a sufficient justification to state that they are not simple to

fabricate. The literature is full of excellent demonstrations of focusing and super-focusing based on “Metas”. Please give some comment on that aspect.

3. Figure 1, the reviewer would like to see a real picture of the sample (Phononic waveguide). A SEM image, for example, could be inserted as a sub-figure in figure 1. It is very important to show the readers the real sample the authors did analyze.

4. The paper is mostly dealing with “temporal focusing” which is very interesting. However, the authors did not give any insight about spatial focusing which is the one of the main properties we look for when we aim any applications regarding wave focusing, lensing and super-focusing. The authors should at least discuss this issue in correlation with the temporal one.

5. The temporal focusing demonstrated by the authors is for 1D elastic wave propagation. Could the authors give some comments on how they can extent this to a 2D configuration ?

General Responses to Reviewer # 3

We are grateful to the reviewer for his/her constructive comments on our study. The reviewer’s suggestion helps us to improve our manuscript.

Point-by-Point Responses to Reviewer #3’s Comments

1. The authors are dealing with elastic waves, not acoustic ones. So they must change this both in the title and in the text.

As the reviewer suggested, we revised the title and the text from acoustic waves to elastic ones.

2. In the introduction, the authors briefly mentioned metamaterials as means for wave focusing. Indeed, metamaterials are presenting some unique properties both for focusing and super-focusing (beyond the diffraction limit). The reviewer wants the authors to emphasize more about the add value of their approach in comparison with metamaterials. Complexity and sophistication of some metamaterials is not a sufficient justification to state that they are not simple to fabricate. The literature is full of excellent demonstrations of focusing and super-focusing based on “Metas”. Please give some comment on that aspect.

We thank the reviewer for his/her critical suggestion. As the reviewer pointed out, acoustic metamaterials show excellent demonstrations of focusing and super-focusing in spatial domain. We recognize the great performance of the acoustic metamaterials. However, most acoustic metamaterials and even phononic crystals demonstrating these unique properties such as nearly diffraction limited focusing and super-focusing rely on locally resonant composite structure and crystal anisotropy of the band structure and this can limit the available operation bandwidth.

On the other hand, our temporal focusing technique can be realised by just adjusting the input chirp parameter and namely, it can be applied to arbitrary frequency. Furthermore, combining this technique with the interesting properties of conventional metamaterials can create new functionality, for instance allowing the temporally-tailored wavepacket e.g. soliton to be two-dimensionally controlled. Thus, we believe that our focusing technique can not only overcome narrow operation bandwidth limitation, but also increase the ability to directionally manipulate the wavepacket by combining it with metamaterials.

Based on the discussion above, we revised Abstract and Introduction parts, and added a reference [R1].

[R1] S. Zhang et al., Focusing ultrasound with an acoustic metamaterial network, Phys. Rev. Lett. **102**, 194301 (2009).

Page 1, Line 9

Although several spatial focusing techniques have been developed, these systems require sophisticated resonant structures with narrow bandwidth, which limit their practical applications.

Page 2, Line 8

Although such structures have been used to demonstrate effects such as nearly diffraction limited focusing and super-focusing in spatial domain, they often require locally resonant structures and crystal anisotropy of the band structure, which results in narrow operation bandwidth and thus limits their practical use.

Page 2, Line 23

Importantly, this temporal focusing method can be applied to an arbitrary dispersive material which imparts quadratic time-varying phase shift. We consider that combining temporal focusing technique with metamaterials which have relatively wide bandwidth increase the ability to manipulate elastic waves.

3. Figure 1, the reviewer would like to see a real picture of the sample (Phononic waveguide). A SEM image, for example, could be inserted as a sub-figure in figure 1. It is very important to show the readers the real sample the authors did analyze.

SEM images of the device are inserted in the right and bottom insets of Fig. 1(a). We revised caption in Fig. 1(a) as follows.

Page 13, Caption

A schematic of the PnC WG and the measurement set-up. The bottom inset shows SEM image of the cross-section of the device which is composed of a GaAs/AlGaAs heterostructure fabricated

by selectively etching the Al_{0.65}Ga_{0.35}As layer, scale bar is 5 μm. The periodic structures are determined by the WG width of 22 μm and the hole pitch of 8 μm as shown right inset (false colored SEM image).

4. The paper is mostly dealing with “temporal focusing” which is very interesting. However, the authors did not give any insight about spatial focusing which is the one of the main properties we look for when we aim any applications regarding wave focusing, lensing and super-focusing. The authors should at least discuss this issue in correlation with the temporal one.

We thank the reviewer for his/her suggestion. While in temporal domain the waveform is Gaussian, in spatial domain the waveform is skewed Gaussian due to group velocity dispersion. We can estimate spatial FWHM (S_{FWHM}) by using the equation E1.

$$U(x, t) = \frac{T_0}{[T_0^2 - ik_2 x(1+iC)]^{1/2}} \exp\left(-\frac{(1+iC)(t-x/v_g)^2}{2[T_0^2 - ik_2 x(1+iC)]}\right) \quad (\text{E1})$$

The width S_{FWHM} of input (leftmost pulse in Fig. 4d) is about 0.48 mm, and the width S_{FWHM} of most compressed pulse is about 0.14 mm from the calculation when the chirp parameter is 9.7. If the device sustains broader bandwidth, the ability to amplify waves is improved. We inserted new section, “S2 Spatial focusing”, in Supplementary Information and inserted the text in “Temporal focusing of an ultrasonic pulse” section as follows.

Page 6, Line 25

where the pulse width is temporally compressed from 1.9 μs to 0.6 μs, and simultaneously, the spatial pulse width to be compressed from 0.48 mm to 0.14 mm (see Supplementary Information). The pulse power is enhanced more than one order of magnitude.

5. The temporal focusing demonstrated by the authors is for 1D elastic wave propagation. Could the authors give some comments on how they can extend this to a 2D configuration?

We thank the reviewer for his/her comments. Recently there has been significant research interest in controlling elastic/acoustic wave propagation by using the crystal anisotropy of 2D band structure. If we expand this temporal focusing technique to the dispersion-engineered 2D metamaterials and phononic crystals, it can be possible to directionally control the temporal wave focusing and expansion, which greatly increases the ability to manipulate elastic waves. We inserted the text in “Discussion” section as follows and added references [R2-4].

Page 9, Line 7

Furthermore, this temporal focusing technique can be applied in two-dimensional phononic crystals and metamaterials where elastic wave propagations are directionally controlled based on the crystal anisotropy of 2D band structures [R2-4], to create new means to manipulate elastic waves on chip

[R2] L. Feng et al., Negative refraction of acoustic waves in two-dimensional sonic crystals, *Phys. Rev. B* **72**, 033108 (2005).

[R3] M. Ke et al., Negative-refraction imaging with two-dimensional phononic crystals, *PRB* **72**, 064306 (2005).

[R4] Y. Guo et al., Acoustic beam splitting at low GHz frequencies in a defect-free phononic crystal, *Appl. Phys. Lett.* **110**, 031904 (2017).

REVIEWERS' COMMENTS:

Reviewer #3 (Remarks to the Author):

The authors have addressed all the points I have pointed out in my review and added all necessary materials to the manuscript to make it relevant and even more accessible to a larger audience. As I said in my first review, this paper produces an original work on temporal focusing of elastic waves based on phononic crystals and presents a real interest for the future development of novel chip based technologies. I then recommend it for publication in NC.

Reviewer #3 (Remarks to the Author):

The authors have addressed all the points I have pointed out in my review and added all necessary materials to the manuscript to make it relevant and even more accessible to a larger audience. As I said in my first review, this paper produces an original work on temporal focusing of elastic waves based on phononic crystals and presents a real interest for the future development of novel chip based technologies. I then recommend it for publication in NC.

Responses to Reviewer # 3

We are grateful to the reviewer for valuing our work and are delighted that the reviewer recommends the publication of this manuscript.